# From Camera Roll to Solid Pod: LLM-Facilitated Privacy-Preserving Data Sharing on Mobile Devices

George Popescu-Craiova

*King's College London, UK*

### Abstract

Privacy risks surrounding personal data use are increasingly acute in data-rich environments such as mobile devices, where large volumes of sensitive data are routinely collected and repurposed for centralized analytics and AI training. Despite growing awareness of these risks, users lack practical, privacy-first mechanisms for interacting with their on-device data and selectively sharing it with federated learning systems or data-sharing platforms. The sheer diversity and scale of personal data, combined with the effort required to manually classify, curate, and manage it according to individual privacy preferences, often leads users to default to coarse-grained or bulk consent. This paper presents an alternative approach in which data classification, user privacy preferences and ongoing data curation are done with a locally deployed large language model acting as a trusted advisor. By combining on-device perception with natural language interaction, users can express nuanced sharing intentions while retaining control over what data leaves their device. We integrate this approach with Solid pods as the data-sharing backend, leveraging their decentralized and user-owned storage model to support fine-grained, auditable, and revocable access control. Together, these components enable a privacy-first data-sharing workflow that avoids reliance on centralized, data-extractive cloud infrastructures.

### Keywords

privacy, Solid, Large Language Models, mobile

## 1. Introduction

The increasing ability to host performant Large Language Models on mobile devices ([1], [2], [3]) allows users to carry over a dialogue over privacy considerations with a trusted, locally hosted LLM that can facilitate data sharing. The rise of LLM-enabled apps ([4]), whether hosted on device or using APIs, means that agentic tooling combined with powerful LLMs can create a new range of AI-assisted user use cases facilitated by natural language. The main driver for adopting a locally hosted LLM deployment is that it's private by design and the data doesn't leave the device, unlike a request to a model hosted on a company server.

This paper proposes a data sharing mechanism that uses LLMs to allow users to curate image sharing based on image classification. Advances in image classification models which are optimised for mobile deployment ([5], [6]) and a general direction towards smaller models such as EfficientNetV2 ([7]) are preferred for optimal image classification tasks that aid the LLM capability in tagging images that users want to share. This works on device by running an async parallel process to tag images into categories. The tagging capability is an attempt at creating categories of data based on images. Given the nature of images that can contain multiple objects, people, screenshots, PII data, children, animals and many different others that users may not want to share, the image classification task will always have an inherent risk of misclassification, which can lead to improper sharing. However, even an imperfect systems is preferred to bulk sharing, which would be the norm in on-device applications that tend to favour bulk image uploads.

Human supervision for the LLM systems remains important, and there are both considerations around data sharing and privacy considerations from a user perspective but also from a data legislation lens. These are baked into the design by instructing the LLM to avoid uploading pictures of PII, pictures

*Solid Symposium 2026: Privacy & Personal Data Management, April 30, 2026, London, UK*

✉ george.popescu-craiova@kcl.ac.uk (G. Popescu-Craiova)

🌐 https://www.georgepopescu-craiova.info (G. Popescu-Craiova)

🆔 0009-0005-5083-4497 (G. Popescu-Craiova)

of children, screenshots and anything else the user might want to exclude. The literature around mobile agentic capability ([8]) shows that LLM capability for high-level tasks in agentic systems is improving and that agentic systems on-device can facilitate tasks on behalf of users. Agentic systems aimed at enforcing privacy standards in data sharing mechanisms on mobile phones are constrained by the deployment environment, whether it is Android or iOS and their considerations for how to interact with systems such as the camera roll. Techniques such as fine-tuning ([9], [10]) the on-device LLMs are desirable as they can incorporate new knowledge pertaining to local user privacy requirements into the system while retaining the training from pre-training.

## 2. Background and Related Work

The management and sharing of personal data has increasingly shifted toward centralized cloud platforms, where user control is limited and data reuse is often opaque. Solid ([11], [12]) proposes a decentralized alternative in which users store their data in personal online data stores (pods) and grant fine-grained access permissions to applications. This model decouples data storage from application logic and provides a foundation for privacy-preserving data ecosystems. Related work ([13]) has proposed data sharing mechanisms that allows authenticated users to segment data granularly. In this paper, Solid is evaluated as a mechanism that could allow granular and sophisticated data sharing mechanisms, in line with the category-based approach from Data Bank [14] which can support granular requests on the model to categorise data and share it on Solid pods.

In parallel, mobile devices have evolved into rich data environments containing images, location traces, communications, and behavioral signals. Prior work has explored on-device machine learning for privacy-sensitive tasks, demonstrating that modern smartphones can host efficient neural networks for inference without transmitting raw data off-device [6, 7]. Image classification and object recognition models optimized for mobile hardware enable coarse semantic labeling of personal media while preserving local data boundaries.

Recent advances in compact Large Language Models (LLMs) have further enabled natural language interaction and reasoning directly on edge devices [1, 3]. Agentic mobile systems demonstrate the feasibility of delegating high-level user intent interpretation to LLMs while executing actions locally [8]. However, most existing work focuses on task automation rather than privacy-centric data governance.

Our work bridges these directions by combining Solid's decentralized storage model with on-device LLM reasoning and lightweight image classification, enabling users to express privacy preferences in natural language while maintaining full control over what data leaves their device.

## 3. System Architecture

The proposed system consists of three primary components operating across the mobile device and the Solid ecosystem: (i) on-device perception and reasoning, (ii) user-controlled decision logic and (iii) privacy-preserving data storage. The purpose of the intended system is to showcase a mechanism for LLM curated granular data sharing over Solid from a data rich source like a user's camera roll.

On-device perception is implemented through a lightweight image object recognition pipeline that scans the mobile camera roll and assigns coarse semantic labels (e.g. flowers, documents, people). This process runs entirely locally on device and produces metadata that is then used by the LLM for categorisation.

Reasoning and decision-making are handled by a locally deployed LLM. The model receives a summarized view of detected data categories along with a user-provided natural language instruction describing their sharing intent. Rather than uploading data automatically, the LLM acts as an advisor that selects candidate data items based on the instruction and previously inferred preferences.

Selected data items are uploaded to a user-owned Solid pod. Applications access this data only through Solid's access control mechanisms, ensuring that data sharing remains auditable, revocable,

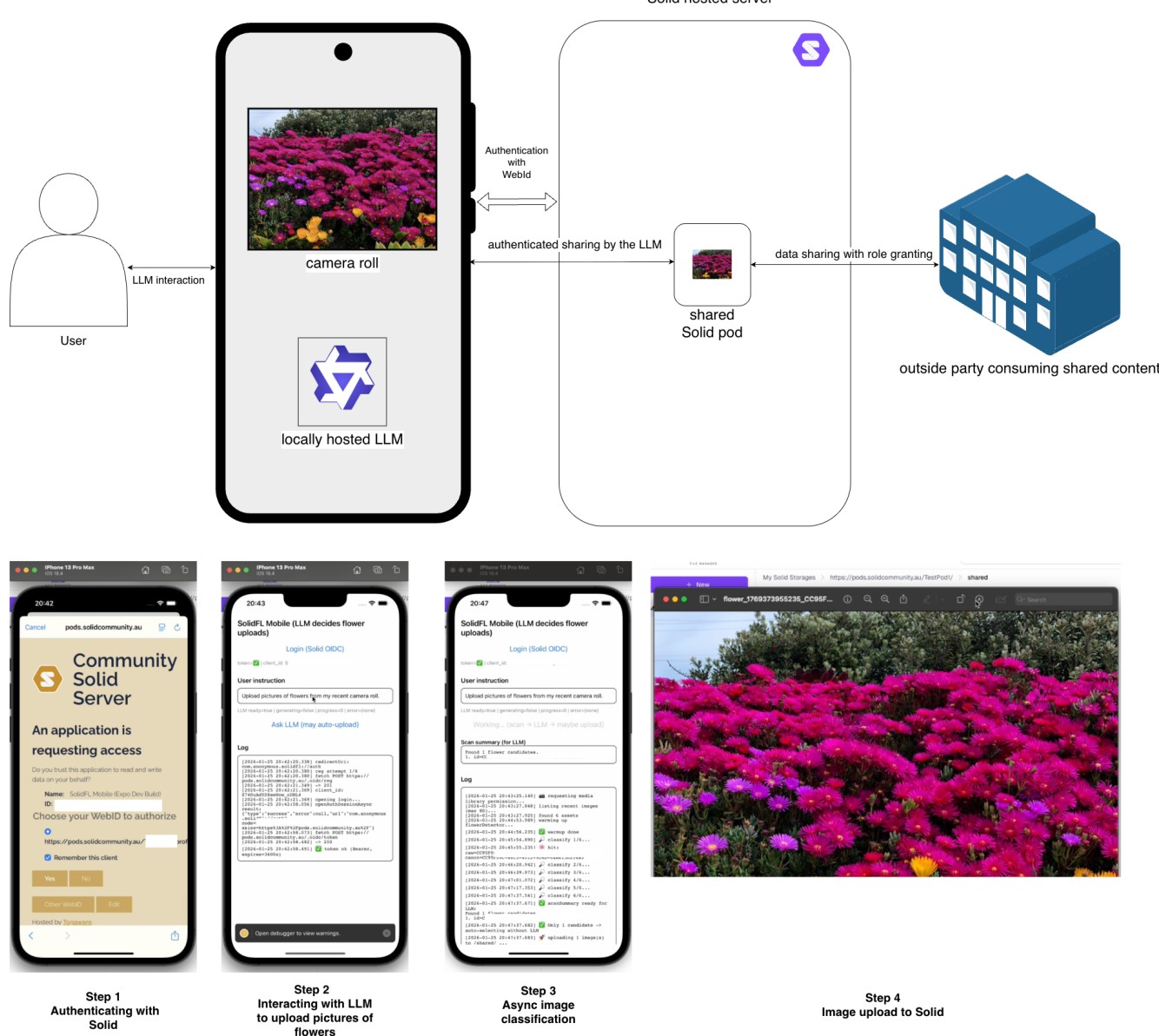

**Figure 1:** System architecture for on-device LLM-guided image sharing over Solid.

and user-defined. No raw data is transmitted to third-party cloud services or any other systems apart from Solid during classification or reasoning.

## 4. Implementation Approach

We implemented a prototype on iOS using React Native and Expo, leveraging platform-supported access to the device media library. Image classification is performed using a MobileNet-style convolutional neural network optimized for on-device inference. Images are resized and processed asynchronously to minimize latency and energy consumption.

The LLM component is deployed locally using a quantized model suitable for mobile hardware. The language model is based on Qwen3-0.6B and fine-tuned using the Unsloth framework [15], following a phone-oriented deployment pipeline. Fine-tuning focuses on aligning the model with privacy-centric behaviors, training it to map natural language instructions and summarized candidate data to structured outputs (JSON arrays of asset identifiers). The training dataset consists of synthetic instruction response

pairs generated to reflect localized, privacy-first decision patterns, emphasizing conservative selection and explicit exclusion criteria. Fine-tuning is performed offline, and only the resulting model weights are deployed to the mobile device. The model is prompted with a structured summary of detected image categories and a constrained instruction format to ensure predictable outputs. Rather than free-form text generation, the LLM produces structured selections of asset identifiers corresponding to images eligible for sharing. This is also needed for being able to interacting with Solid in a programmatic fashion.

Uploading is explicitly triggered only after LLM selection, ensuring that perception and decision-making remain decoupled from data transmission. Files are uploaded directly from local storage to the Solid pod using authenticated HTTP PUT requests, avoiding intermediary servers.

This modular design allows individual components to be replaced or extended without altering the trust boundaries of the system.

## 5. Privacy Guarantees and Verification

The system provides privacy guarantees through architectural separation and local on-device privacy. All raw data, model training, image classification and inspection and language reasoning, occurs on the user's device. Only data explicitly selected by the user-facing LLM is transmitted externally.

Solid's access control mechanisms ensure that uploaded data is available only to authorized agents and can be revoked at any time. Since data is stored in user-controlled pods, applications cannot silently retain or repurpose data beyond granted permissions.

Potential misclassification by image recognition models is mitigated by conservative defaults and user oversight. The system favors under-sharing rather than bulk uploads, significantly reducing exposure compared to centralized photo backup services.

Verification of privacy properties is supported through transparent logs of classification decisions, LLM outputs and upload actions, enabling user inspection and auditing.

## 6. Discussion

This work demonstrates how LLMs on mobile devices can act as active privacy mediators rather and transform data-rich ecosystems into dynamic and easy to curate data sources. By combining local perception with language-based reasoning, users can express nuanced sharing preferences without manual data curation.

Solid pods provide a natural storage backend for such systems, as they align with the principle of user data sovereignty. The integration of LLMs further reduces the cognitive burden associated with managing large personal datasets. Solid allows a permissions to be established at a pod, container, or file level and can allow read, write, append, or control permissions to entities with a WebID through the Access Control Policy [16] as the policy language. Using this mechanism, access to resources can be done granularly if objects such as images are segmented through metadata into multiple levels each with different sets of permissions for different agents.

More broadly, this approach suggests a shift away from consent-through-obscurity toward interactive, intent-driven privacy management, where users engage in ongoing dialogue with their devices about data use.

## 7. Limitations and Future Work

The current prototype relies on coarse image categories and does not perform fine-grained content filtering such as face recognition or sensitive attribute detection. Classification errors remain possible and may lead to unintended exclusions or inclusions.

LLM behavior is constrained but not formally verified, and future work should explore methods for validating decision consistency and robustness. Energy consumption and performance trade-offs on lower-end devices also warrant further evaluation.

Future extensions include multi-modal reasoning over additional data types, personalized privacy preference learning, and integration with federated learning workflows that respect Solid-based access controls.

A mechanism that explores granular access using Solid's access control mechanism and includes data retrieval mechanisms can showcase how a mature data-sharing ecosystem around Solid can evolve.

## 8. Conclusion

We presented a privacy-driven data sharing approach that combines on-device LLM reasoning, mobile image classification, and Solid pods. By keeping perception and decision-making local while using decentralized storage for sharing, the system empowers users to control their data through natural language interaction.

This work illustrates how emerging mobile AI capabilities can support privacy-first architectures and offers a practical step toward more transparent, user-centered data ecosystems.

## Acknowledgments

Many thanks to Maribel Fernandez for helping with the paper review.

## Declaration on Generative AI

During the preparation of this work, the author(s) used ChatGPT5.2 for grammar and spell checking. After using these tool(s)/service(s), the author(s) reviewed and edited the content as needed and take(s) full responsibility for the publication's content.

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

## A.  Online Resources

The code base for this implementation can be found at:

- GitHub