# OpenReview forum: "From Camera Roll to Solid Pod: LLM-Facilitated Privacy-Preserving Data Sharing on Mobile Devices"
_SolidProject.org/SoSy/2026/Privacy_Session — SoSy2026-Privacy Paper_

### Official Review · ~Patrick_Hochstenbach1 · 2026-02-24
**Data Sharing Between Mobile Devices Using Image Classifiers and an On-Device LLM**

**Rating:** 6
**Confidence:** 5

**Review:**

The paper proposes a data‑sharing mechanism that seeks to avoid the coarse‑grained or bulk consent often encountered on mobile devices when sharing camera‑roll content with cloud systems. Instead, it suggests using AI technologies in combination with human supervision to select camera‑roll images that may be uploaded to a personal Solid Pod. While no further details about the requirements for the Solid environment are provided, it is assumed that users can further manage access permissions to control which parties may interact with the shared content.

The paper presents a demonstrator showing how users can employ natural language to instruct an on-device agent to upload images from a mobile device’s camera roll to a Solid Pod. The demonstrator includes an on‑device image classification and object recognition model, a compact on‑device Large Language Model (LLM), and an administrative application that allows users to specify which types of images to upload, view the image classifications, and review the log of LLM and Solid interactions.

The text is clearly written, however some definitions of acronyms are nor provided (such as PII) or too late in the text (LLM).

The quality of this contribution is in providing a demonstrator for running image classificators and LLMs on device and providing an administrative interface for uploading content to a Solid pod. However, the description of the solution is limited to qualitative explanations. The introduction states that an imperfect bulk sharing is preferred to bulk sharing into the cloud. Still some numerical values would provide insight of the state of the art that this paper presents for the combination of on-device image classification and LLM interactions.

It is unclear in this paper how consent is defined. Is the argument that the user consents to an LLM making the choices what to share? Is the argument that the user still decides what to share but uses an LLM to provide an easier way to filter the camera roll?

Some assumptions made about Solid Pods are not correct. For example, the statement “Since data is stored in user‑controlled Pods, applications cannot silently retain or repurpose data beyond granted permissions” is inaccurate or wrong (depending on the meaning of repurpose/permissions). Once read access is granted to an external party, the Pod cannot control how that party may repurpose the data. What is required is a policy that explains the purpose of data sharing, as well as the legal obligations associated with sharing the data. This is the consent that is required. The paper does not mention these policies or legal requirements.

The discussion section provides too many details about Solid, which could be omitted in favour of offering more information about the demonstrator and the envisioned approaches to addressing the consent‑for‑sharing problem. It would also have been useful for the author to pose questions to the community about challenges encountered during the project, which could stimulate discussion at the workshop.

The argumentation that local = secure also needs some further discussion as there are many ways how mobile devices can get compromised.

Side remark: the paper mentioned that the code base of the implementation can be found on GitHub. On February 24 2026 it contained the screen recording of the solution, not the code base.

Side remark: figure 1 would be clearer if the locally hosted image classifier would be included

---

### Official Review · ~Muhammad_Raza_Naqvi1 · 2026-03-02
**The work is borderline in terms of calrity and novelty. While it presents results, the manuscript lacks sufficient experimental evaluation, transparency, and reproducibility to fully support its claims. Substantial improvements—such as accessible code, proper evaluation on multiple examples, clear discussion of limitations, and adherence to FAIR principles—are needed for this to constitute a robust, publishable contribution.**

**Rating:** 5
**Confidence:** 5

**Review:**

-----significance of this work

The manuscript proposes using a locally deployed large language model as a “trusted advisor” for data classification, user privacy preferences, and ongoing data curation. This raises significant concerns, as it is unclear how the model can reliably be considered trustworthy. Furthermore, the image classification task is based on some images, which limits the evaluation of the approach and makes it difficult to assess the model’s effectiveness, robustness, or suitability for practical use.

Even when deployed locally, large language models can behave unpredictably. For instance, if a locally deployed LLM is used for image classification or privacy decisions, subtle biases in the model’s training data could lead to misclassifications or privacy violations. Unlike traditional deterministic algorithms, LLMs may produce inconsistent outputs on similar inputs, and there is no inherent mechanism to audit or verify every decision. This unpredictability makes it difficult to trust the system, especially in critical applications involving user data or automated decision-making.

 In particular, it is unclear why Qwen 3–0B was chosen and fine-tuned using the Unsolt framework, and no justification or comparative analysis is provided. Discussing these aspects and reporting negative results would strengthen the work by providing a more complete and transparent evaluation.

---- Clarity

The claim that perception and reasoning occur on-device via a locally deployed LLM is ambitious. LLMs are not inherently reliable for any time of reasoning, and without evidence or safeguards, it is difficult to judge the system’s accuracy, consistency, or trustworthiness.

The author mentions in the manuscript that the system handles semantic labels such as “flowers,” “people,” and documents. However, the provided video only shows examples of mountains and flowers, with no demonstration of “people” or other mentioned labels. This discrepancy makes it difficult to evaluate the system’s claimed capabilities and raises questions about the completeness of the demonstration.

--Quality

The manuscript frequently cites non–peer-reviewed or tangential sources more than 7, and some references appear irrelevant to the claims being made. This weakens the credibility of the arguments and highlights the need to rely on peer-reviewed, directly relevant literature.

Additionally, the manuscript shows inconsistent citation formatting: it begins using ([1]) and then, after reference 13, switches to []. This inconsistency is not good practice and undermines the quality of the paper.

Author stated that code base to be released publicly after cleanup and documentation. However, as no code is currently accessible, the repository adds little value beyond what is already presented in the paper’s figure such as Fig.1. The system description does not follow FAIR principles, since key information about its functioning is not shared. Without accessible code or additional details, the GitHub repository does not meaningfully demonstrate the system’s applicability or support reproducibility.

---

### Decision · Program_Chairs · 2026-03-09

Accept (Paper)